# Effect of α-Lipoic Acid on the Development of Human Skin Equivalents Using a Pumpless Skin-on-a-Chip Model

**DOI:** 10.3390/ijms22042160

**Published:** 2021-02-22

**Authors:** Kyunghee Kim, Jisue Kim, Hyoungseob Kim, Gun Yong Sung

**Affiliations:** 1Interdisciplinary Program of Nano-Medical Device Engineering, Graduate School, Hallym University, Chuncheon 24252, Korea; seoulhee92@naver.com (K.K.); prtty_u5588@naver.com (J.K.); hyoungseob12@naver.com (H.K.); 2Integrative Materials Research Institute, Hallym University, Chuncheon 24252, Korea; 3Major in Materials Science and Engineering, School of Future Convergence, Hallym University, Chuncheon 24252, Korea

**Keywords:** pumpless skin-on-a-chip, α-lipoic acid, human skin equivalents, skin anti-aging

## Abstract

Owing to the prohibition of cosmetic animal testing, various attempts have recently been made using skin-on-a-chip (SOC) technology as a replacement for animal testing. Previously, we reported the development of a pumpless SOC capable of drug testing with a simple drive using the principle that the medium flows along the channel by gravity when the chip is tilted using a microfluidic channel. In this study, using pumpless SOC, instead of drug testing at the single-cell level, we evaluated the efficacy of α-lipoic acid (ALA), which is known as an anti-aging substance in skin equivalents, for skin tissue and epidermal structure formation. The expression of proteins and changes in genotyping were compared and evaluated. Hematoxylin and eosin staining for histological analysis showed a difference in the activity of fibroblasts in the dermis layer with respect to the presence or absence of ALA. We observed that the epidermis layer became increasingly prominent as the culture period was extended by treatment with 10 μM ALA. The expression of epidermal structural proteins of filaggrin, involucrin, keratin 10, and collagen IV increased because of the effect of ALA. Changes in the epidermis layer were noticeable after the ALA treatment. As a result of aging, damage to the skin-barrier function and structural integrity is reduced, indicating that ALA has an anti-aging effect. We performed a gene analysis of filaggrin, involucrin, keratin 10, integrin, and collagen I genes in ALA-treated human skin equivalents, which indicated an increase in filaggrin gene expression after ALA treatment. These results indicate that pumpless SOC can be used as an in vitro skin model similar to human skin, protein and gene expression can be analyzed, and it can be used for functional drug tests of cosmetic materials in the future. This technology is expected to contribute to the development of skin disease models.

## 1. Introduction

The average age of the population has gradually increased worldwide. Aging results in increased skin sagging, wrinkles, and cutis laxa [1]. Aged skin has been reported to exhibit impaired barrier function, dryness, and an increased risk of skin disease [2]. Additionally, the risk of malignant tumors is increased [3]. A system for the development of anti-aging substances is needed to delay skin aging and reduce the harmful effects of aging.

Skin aging is induced by both intrinsic and extrinsic factors [4,5], which reduce the structural integrity of all tissues and cause a loss of physiological function [5]. Endogenous skin aging is associated with decreased cellular replication capacity and increased degradation of the extracellular matrix. The replication capacity of all dividing cells diminishes over time. In the skin, this process is called cellular senescence, which affects keratinocytes (KCs), fibroblasts (FBs), and melanocytes [6]. Reactive oxygen species (ROS) play an important role in skin aging. Approximately 1.5–5% of the oxygen consumed by the skin is converted to ROS through an implicit process [7]. ROS are continuously produced as a by-product in the electron transport chain of aerobic metabolism in the mitochondria and are considered a major cause of endogenous aging. KCs and FBs are the major producers of “mitochondrial” ROS in the skin [4]. Exogenous aging is caused by environmental oxidative factors, and skin aging caused by extrinsic factors is different from aging of other tissues [4,8].

For preventing aging, it is important to prevent the oxidative action that occurs in tissues through an antioxidant action. In our body, antioxidants produce water and carbon dioxide during the process of energy generation in the mitochondria. During this process, several molecules move and exchange energy, producing considerable energy and leaving a residue that causes oxidation or death of cells if mitochondrial defects are present [9]. An antioxidant component is needed under these conditions. The five major antioxidant components that produce a major effect on our body are α-lipoic acid (ALA), glutathione, vitamin C, vitamin E, and CoQ10.

Removing the active ingredient that accelerates aging of the body involves the donation of one electron and the oxidation of the electron donor. Subsequently, the component that interacts with this component donates an electron as well, and oxidation of the electron donor occurs continuously. For example, vitamin C donates an electron and is oxidized to remove the oxidative stress active component of cells, and ALA replenishes deficient antioxidant components through the process of reduction vitamin C again to prevent aging [10,11].

ALA and dihydroxyacetic acid can directly remove (neutralize) ROS and reactive nitrogen species (RNS) [12]. Metal chelation prevents the formation of free radicals and oxidative damage [13]. ALA can increase glutathione synthesis in aged mice [14,15]. ALA increases the expression of γ-glutamyl cysteine ligase (γ-GCL), which is a restriction enzyme in glutathione synthesis, and the cellular absorption of amino acids required for synthesis; it was reported that the expression of γ-GCL and other antioxidant enzymes was up-regulated via activation of nuclear factor E2 related factor 2 (Nrf2)-dependent pathway [16,17]. ALA is a powerful antioxidant because it plays a large role in this process.

Several recent studies have evaluated the aging effect in vitro. This is a trend toward banning recent animal testing. The European Union passed the total ban on cosmetic animal testing in 2013; approximately four years later, animal testing for cosmetics was banned in South Korea. This has increased interest in alternative methods to animal testing. These methods do not use animals, or they use approaches that can reduce the number of animals or reduce distress (3Rs; replacement, reduction, and refinement) [18]. Alternative methods for evaluating skin irritation have been advanced through the use of a skin model that mimics the characteristics of the human skin [19]. Existing animal models are structurally different from the human skin; additionally, experiments on these models involve animal cruelty, and they are expensive. Furthermore, their immune responses are often different from immune responses in humans, leading to the problem of interspecies differences [20]. Alternative methods to animal testing are required because 2D cell-culture models do not mimic the complicated 3D environment in the living body, and cell analysis using these models can lead to inaccurate results regarding drug reactions [21,22]. To overcome these limitations, a 3D cell-culture model in which cells are cultured using ECM scaffolds is being studied. 3D culture has the advantage of exhibiting complex biological characteristics [23,24]. As an alternative method to animal testing, biochip technology with 3D culture skin-on-a-chip (SOC) utilizes skin equivalents with characteristics similar to those of the human skin for evaluating the skin toxicity of cosmetic ingredients and drug screening. The evaluation can proceed quickly and easily. SOC technology can be used to test the skin’s response to chemical substances [25], drugs [26], and UV irradiation [27]. SOC was developed by utilizing several layers of cell cultures [26,27,28] or 3D skin equivalents [25,26]. In recent years, various attempts have been made, such as dendritic cells [27], vascular channel [28], and nonshrinkable fiber-based matrix.

In this study, we applied SOC technology to utilize a pumpless skin-on-a-chip, which enables easy operation and drug testing. For the circulation of the medium, which is essential during culturing when using SOC technology, we created a microfluidic chip that does not require a pump so that can be driven more easily than a system driven by an existing pump. Previous SOC culture systems exhibited the disadvantage of complicated device configuration, which required additional pumps to generate fluid flow [29,30] and additional tubing for that purpose. Pumpless SOC uses a microfluidic chip that applies the principle that gravity causes the medium to flow along the microfluidic channel when the chip is tilted [31]. It is possible to create a biomimetic environment that enables various physiological functions without a pump, such as supplying nutrients to cells via microfluidic channels and removing cell waste. The pumpless skin-on-a-chip can be designed to have a physiological residence time that matches the blood flow in the tissue to obtain a drug-related concentration profile in the blood. It can be driven without a pump or external tube, making it simple and free from tube contamination. It is a more convenient system without the drawback of having to rely on a drive pump and without the problems caused by air bubbles, which is very problematic in tube experiments. Three type I collagen studies have confirmed that rat-tail collagen is the most appropriate scaffold [32] and that pumpless SOC produces anticancer-drug-induced side effects on the skin [33] and anti-aging effects. We have reported the effects of curcuma longa leaf extract (CLLE) [34] and CoQ10 drugs [35].

In this study, rather than previously reported single-cell level for drug tests, we evaluated the possibility of using skin tissue formation, epidermal structural protein expression, and gene expression to develop a skin substitute model.

## 2. Results

### 2.1. Changes in Contraction of Full-Thickness Human Skin Equivalents (HSEs)

Contraction of cell cultures is displayed by organizing the cells in a 3D culture. Cell-mediated collagen contraction occurs in 3D culture, which is known to be similar to the process that occurs when cells interact with protein fibers and collagen in 3D culture [36]. It seems clear that the contraction phenomenon appears depending on the activity of fibroblasts [36]. Paracrine factors secreted by dermal fibroblasts fine-tune the balance between keratinocyte proliferation, migration, and differentiation via dual paracrine signals when ball-cultured with keratinocytes [37]. It appears to affect contractions such as ECM synthesis and dermal remodeling [38]. The contraction of the collagen scaffold can be used as an assay to determine whether the cocultured tissue is properly cultured. Shrinkage rate is measured as an indicator that can monitor a real-time sample of fibroblasts-collagen (the reason for the experiment, no contraction when cells die) corresponding to the dermis layer of cultured tissue with no cell activity or reproducibility. We investigated the contraction of HSEs cultured via pumpless skin-on-a-chip during the culture period using photographs taken at 24 h intervals. The differences between the experimental group treated with ALA and the control group, which was not treated with ALA, were compared (Figure 1). Figure 1 presents the differences in contraction during the culture period. The dermis layer (DL) was formed during the 5 day period after the start of the culture, and the epidermal layer (EDL) with FBs was formed during day 5 to day 7. Additionally, we established a coculture of KCs; ALA was cultured in groups treated for 3, 5, and 7 days during an air exposure (AE) period of 8 to 14 days.

Figure 1 indicates that there was almost no contraction in the groups treated with ALA for 3 days (Figure 1a) and 5 days (Figure 1b) during day 0 to day after the start of the culture. Only the group that was treated with ALA for 7 days (Figure 1c) exhibited a contraction of approximately 9.35%. This value was small compared to the final contraction rate, 70%, so it did not mean much.

Although the contraction rate showed a radical increase 5 to 7 days after the start of the culture, the group treated with ALA for 3 days (Figure 1a) showed a difference between the samples. The control group (■), which was treated with 0 μM ALA (Figure 1a), exhibited an average contraction of 51.52%. The groups scheduled to be treated with 1 μM (●) and 10 μM (▲) ALA showed an average contraction of 35.39% and 31.06%, respectively.

AE was performed, with the control exhibiting the most contraction in the group treated for 3 days. Figure 1b indicates a difference between the samples, even in the group treated with ALA for 5 days. The average contraction rate was 45.99%, 54.61%, and 39.33% in the group treated with 0 μM (control group; ■), 1 μM (●), and 10 μM ALA (▲). ALA treatment was performed for 5 days (Figure 1b) on the sample that showed the least contraction in the group treated with 10 μM ALA. Figure 1c indicates a similar contraction among samples in the group treated with ALA for 7 days. The average contraction rate in the ALA 7-day treatment group (Figure 1c) was 53.38%, 55.66%, and 54.41% for the group treated with 0 μM (control group; ■), 1 μM (●), and 10 μM ALA (▲). All groups subjected to a 7-day ALA treatment with AE exhibited similar contractions.

Because the contraction rate during day 0 to day 7 after starting the 3D culture from the chip was recorded before ALA treatment, it is possible to compare the treatment effect of ALA accurately over day 8 to day 14 (Figure 1d). The difference in contraction rate can be confirmed by comparing the degree of contraction. The change in contraction rate during ALA treatment (%) reflected the change in contraction rate between the day 7 of the culture during the EDL-formation period and the ALA-treatment period. Since it is a measurement of shrinkage on the photo, there may be some measurement error on some days. The data in Figure 1d were averaged by increasing the number of n to 4 or more to reduce this. We observed that the contraction rate (%) of the group (Figure 1a) treated with ALA for 3 days changed from 51.52% to 62.15% for the control (■). During this period, the contraction rate increased or decreased by 20.65%. The daily change in contraction (percent per day) was 3.55%. The increase or decrease in contraction rate (%) for the 1 μM (●) treatment group was 57.27% (35.39% → 55.66%), and the daily change in contraction (percent per day) was 6.76%. The increase or decrease in contraction rate (%) for the 10 μM (▲) treatment group was 117.17% (31.06% → 67.46%), and the daily change in contraction (percent per day) was 12.13%. As shown in Figure 1b, the increase or decrease in contraction rate (%) for each experimental group in the group treated with ALA for 5 days was 26.55%, 28.70%, and 68.29% for the control (■), 1 μM (●), and 10 μM (▲), respectively. The daily changes in contraction of the control (■), 1 μM (●), and 10 μM (▲) (percent per day) were 2.44, 3.13, and 5.37, respectively. As shown in Figure 1c,d, increase or decrease in the contraction rate (%) for the control (■), 1 μM (●), and 10 μM (▲) groups in the group treated with ALA for 7 days were 29.12%, 36.70%, and 41.41%, respectively; the daily changes in contraction (percent per day) were 2.22%, 2.92%, and 3.22%, respectively.

### 2.2. Examination of Histological Changes in ALA-Treated HSEs

HSEs produced by the pumpless skin-on-a-chip are designed to react similarly to real human skin in a full-thickness model composed of the dermis and epidermis layers. The dermis layer was constructed with human fibroblasts (HFBs) using 6.12 mg/mL rat-tail collagen (RTC) in the extracellular matrix (ECM). In this study, ALA treatment was performed at 0, 1, and 10 μM during the AE period, which is the stage of inducing differentiation, and comparative analysis was performed during the culture period under treatment with the three concentrations.

Figure 2 summarizes the changes in the histological structure, as evaluated using hematoxylin and eosin (H&E) staining. In the group treated with ALA for 3 days, when compared with the control (Figure 2a), treatment with 10 μM ALA (Figure 2c) resulted in the formation of a thicker epidermis and the most remarkable differentiation of the stratum corneum. In the group treated with ALA for 5 days (Figure 2d–f), no difference was observed in the thickness of the stratum corneum formed, with respect to ALA concentration. However, there were differences in the number and shape of FBs cells observed in the dermis layer depending on the presence or absence of α-lipoic acid. Many elongated cells were observed that could be inferred to have high activity of most FBs in the ALA-treated group, and more fiber morphology was observed around the FBs.

The cell of the dermis layer was manually counted using the cell counter plugin of the Image J program. Seven days 0 μM was 38, 1 μM was 38, and 10 μM was 50. (Number of fibroblast cells in a 20x image of skin equivalents) When treated for 7 days, the number of cells forming the dermis at 10 μM was relatively larger than at 0 μM and/or 1 μM. Regarding the changes in tissue morphogenesis according to the culture period in the control group, a longer culture period resulted in a more prominent epidermis layer formation and an increase in number and activity of FBs in the dermis layer.

On the basis of the H&E staining in Figure 2, we confirmed that ALA processing was affected. The expression of representative proteins constituting the epidermis and dermis layer was confirmed through 3,3- diaminobenzidine (DAB) immunohistochemistry (IHC) staining.

Considering the results of IHC staining of filaggrin, as presented in Figure 3, there was a clear difference between the 0 μM control group and ALA-treated HSE group at day 3. In the ALA-treated HSEs, the thickness of the epidermis layer was relatively thick, and the expression area and expression intensity of filaggrin were high. In the group on the third day, the culture period remained short, and epidermis formation was thin. On the 7th day, the thickest epidermis formation was observed, and the filaggrin region was wide and well formed. Through filtering and quantifying only DAB staining using ImageJ, we observed that the group on the 5th day exhibited the highest protein expression. On days 3 and 7, filaggrin expression was relatively high in the ALA-treated group. HSEs treated with ALA for 7 days showed 1.5 times higher expression than the control. There was a significant difference (*p* < 0.05) between the untreated group at 1 μM at day 3 and at 10 μM at day 5.

As shown in Figure 4, increased involucrin expression was observed in HSEs treated with ALA. The longer the ALA treatment period, the higher the amount was of involucrin expressed over a larger area. The group on the 7th day exhibited a high expression of involucrin in the largest area. In the group cultured for 3 days, ALA-treated HSEs exhibited 1.9- to 2-fold or greater expression compared to the control. In HSEs cultured for 7 days, when treated with 10 μM ALA, the expression was 1.95 times higher than that in the control, and a significant difference (*p* < 0.01) was observed.

However, loricrin showed overall and stained results, as shown in Figure 5. Using ImageJ, filtering was performed according to the range of high expression intensity, and only the DAB-staining intensity in the region where the target protein was expressed was confirmed. According to the quantification of DAB staining in the target portion, an increase in expression due to ALA treatment was observed from the 3rd day. In addition, decreased loricrin expression was observed in the ALA-treated group. In particular, a sharp decrease was observed from the 5th day. There was a significant difference (*p* < 0.05) between HSEs treated with 10 μM ALA at day 3 and the untreated control group.

The staining results for keratin 10, as presented in Figure 6, showed a proportional increase in ALA-treated HSEs in the group at day 3 as the concentration increased. The control group, with 0 μM ALA-treated HSEs, showed an increase in stratum corneum thickness and an increase in the expression rate of keratin 10 during the culture period. In the group at day 5, there was no expression of keratin 10 when treated with ALA, and it appeared lower than that in the control. In the day 7 group, a low expression was observed only at 1 μM, with the highest level of keratin 10 protein expression in 10 μM ALA-treated HSEs.

Figure 7 indicates that collagen IV was highly expressed in the dermis and that the expression of collagen IV was high in the ALA-treated groups at days 5 and 7, excluding the 1 μM ALA-treated group at day 5. Even when only DAB was quantified using ImageJ, HSEs treated with 10 μM ALA at day 5 showed a 2.99-fold higher expression than the control, indicating a significant difference between the two conditions. In particular, it was confirmed that ALA-treated HSEs on the 7th day showed 5.19-fold (1 μM) and 4.83-fold (10 μM) higher expression than the control. Additionally, IHC-stained images showed that ALA-treated HSEs at days 5 and 7 showed relatively high expression at the interface of the basal layer.

### 2.3. q-PCR Results for Quantitative Analysis

Relative quantification was performed by comparing the gene expression levels with respect to the difference in the ALA-treatment concentration among the ALA-treated groups. Because the throughput and culture period of ALA-treated HSEs are different in all experimental groups, it is assumed that there are differences depending on the culture period and drug treatment conditions. As a control for the relative quantification of qPCR, the 0 μM condition on the third day, which had the shortest culture period and was not affected by ALA, was used. Since KCs react relatively sensitively to ALA, as confirmed through advanced histological staining, the results were analyzed with an emphasis on the epidermis layer. We analyzed changes in the expression of the involucrin (IVL), ITGB1 (integrin beta1), filaggrin (FLG), and KRT10 (keratin 10) genes, which are the major genes expressed in the epidermis. We compared COL1A (collagen I), which is an important gene for ECM formation in the dermis.

Considering the FLG gene expression in Figure 8a, there was an increase in FLG gene expression with the culture period with 0 μM ALA treatment; however, in the ALA-treated group, the overall filaggrin expression tended to decrease as the culture period increased. The ALA-treated group showed a higher expression than the control group (0 μM). The difference was the greatest in the group on the third day. In the 7th day group, the expression was the lowest with 10 μM ALA treatment. Thus, ALA treatment influences increased expression of the FLG gene.

The expression of the IVL gene tended to decrease as culture period increased in the 0 μM ALA-treatment group (Figure 8b). Consequently, the IVL protein level tended to decrease, overall, with a longer culture period. During ALA treatment, when 1 μM treatment was performed in each culture-period group, relatively high levels of expression were exhibited in the groups at days 3 and 5.

As shown in Figure 8c, the KRT10 gene promotes the expression of differentiation biomarkers and plays a role in maintaining skin homeostasis. In particular, it is affected by the formation of the stratum corneum cell membrane. KRT10 was expected to produce an effect similar to that of the IVL gene, because KRT10 is expressed in large amounts through which a large amount of keratinous cell membrane is formed. Expression appears to be similar to the IVL gene expression phenomenon presented in Figure 8b. The increase in gene expression after ALA treatment was the highest in the group at day 3 after treatment with 1 μM ALA. This is the same phenomenon observed for IVL. In general, when ALA treatment was performed, the expression was lower than that in the control.

As shown in Figure 8d, the expression of ITGB1 showed a sharp decrease at day 7 in the control group, which was treated with 0 μM. Even in the group treated with ALA, we observed a low level of expression on the 7th day. Since this group was compared with the sample on the 3rd day, it is considered that the expression of the ITG gene decreased as the culture period increased. Additionally, the change with and without ALA treatment was the lowest in the group treated with a high concentration of ALA.

COL1A gene expression was performed to compare the change in ECM around FBs due to the increased expression of FBs, as observed by H&E staining, and the change in the expression of collagen that adheres to the epidermis. The COL1A gene expression (Figure 8e) was particularly high in the control group from day 5, and it decreased at day 7. Regarding ALA treatment, increased expression of COL1A appeared in the experimental group treated with 1 μM on the 3rd day. The expression of COL1A tended to decrease overall.

## 3. Discussion

### 3.1. Changes in Contraction of Full-Thickness HSEs

The contraction of the culture can be used as an index of cell activity and culture conditions. It is presumed that if cell activity, cell division, and differentiation are stimulated, the scaffold supporting the cell would contract while being affected by changes in the state of the cells. Therefore, we hypothesized that the contraction rate could be used as a cell-condition index in 3D culture. To confirm this theory, we monitored the contraction of HSEs during the culture period. As shown in Figure 1, the contraction rate when ALA treatment was performed increased in all groups compared to that in the control group. The contraction rate increased with increase in concentration from 1 to 10 μM. We confirmed that the change in HSE contraction progressed more rapidly as the ALA-treatment concentration increased. Overall, the daily change in contraction (percentage per day) tended to decrease as the AE period increased. This indicates that when SEs reach a certain level even when they contract, the contraction slows down or stops. Contraction of HSEs occurs because of the formation of stronger tissues. The results indicated that physiological phenomena such as division and differentiation of constituent cells for tissue formation occur more actively and frequently as the frequency of the contractions increases. It can be found using the results that physiological phenomena such as division and differentiation of constituent cells for tissue formation by the paracrine signaling of fibroblasts occur so actively that more contractions occur. When measuring the contraction rate in real time, if there is a phenomenon that unique contraction does not occur under the same conditions of HSEs, it is considered a “bad” experiment, and it is inferred that there is an experimental problem in the experiment. On the basis of these results, we consider that the contraction rate can be monitored as an index of the physiological activity of HSEs.

### 3.2. Effect of ALA Treatment on Histological Changes in HSEs

Among several concentrations of ALA, we selected the ALA concentration at which cell death was not induced. Although these results have not been presented in this article, the activity of FBs was observed in the concentration range of 10 µM or higher; however, cell death of KCs occurred in this concentration range. Because KCs are primary cells and highly sensitive, the cell death may be attributed to the amount of ALA that needed to be processed. The fact that FBs show activity does not indicate that high concentrations of ALA are unconditionally bad; however, because the state of KCs is important for constructing HSEs, we proceeded with the study by using ALA concentrations of 10 μM or less. Severe culture conditions for KCs have been reported in several studies [39,40]. In this study, the medium used for coculture was prepared and used according to the conditions for KC culture. From the results of H&E in Figure 2, in the case of the ALA-treated group compared to the control, a relatively clear differentiation was seen in the epidermis, and it was found that the dermis was also formed into an elongated shape showing activity, which could better show the network between FBs.

The untreated control groups were compared with the ALA-treated groups. ALA was treated for 3, 5, and 7 days, and the temporal change with the culture period was compared. The H&E staining was analyzed with an emphasis on the epidermis layer, as KCs respond more sensitively to ALA. The results confirmed that for the 10 μM treatment, the differentiation of each layer constituting the stratum corneum proceeded smoothly compared to that in the control. Treatment with 10 μM ALA for 7 days induced the cellular activity of FBs. In the formation of the epidermis layer, the KCs of the stratum basale move upward, forming corneocytes. Corneocytes, which are anucleate cells, fall off as they become keratinized, forming various proteins and lipids. Through this process, the epidermal layer constitutes four major layers and maintains the epidermal layer of a certain thickness [41,42]. It can be inferred that the longer the culture period until the 7th day, the higher the differentiation activity and the thicker the stratum corneum. Additionally, the H&E staining results confirmed that the longer the culture period, the more strongly the HSEs induce the formation of mature tissue. Accordingly, when examining the effect of the treatment period of ALA, the result for the 10 μM ALA-treatment group on the 3rd day and that for the control group on the 7th day were regarded as similar. This phenomenon is believed to be involved in the mechanism through which ALA contributes to the division and differentiation of KCs to form the epidermis. Thus, ALA, as an antioxidant, helps maintain the skin composition. ALA may help protect the skin, such as by forming a skin barrier in the development of the stratum corneum, regulating water and nutrients, and preventing the penetration of external factors such as bacteria and viruses.

### 3.3. Effect of ALA Treatment on Protein Expression of HSEs

IHC DAB-staining intensity was quantified using ImageJ. The epidermis of the skin is differentiated to form the stratum corneum through programmed cell death, which is characteristic of KCs. As epidermal cell proliferation, differentiation, and death progress in chronological order, the expression of specific proteins occurs through each process [42,43].

Keratohyalin granules are a product of KC differentiation in the stratum granulosum. In keratohyalin granules, filaggrin exists in the form of profilaggrin, which is hydrolyzed by proteolytic enzymes as a precursor to filaggrin protein through dephosphorylation. Filaggrin adheres to keratin as a filament-aggregating protein, and it is involved in the binding of cornified cell envelope proteins (involucrin and loricrin) [44,45]. The expression of ALA was examined using IHC staining for investigating the effect of ALA on the expression of filaggrin in HSEs. The expression level of filaggrin did not decrease in HSEs treated with ALA; it was maintained at a constant level. HSEs treated with ALA for 7 days exhibited an expression level of 1.5 times or more compared to that of the control group. This enabled the activation of ALA in the process of KC differentiation without compromising the efficiency of profilaggrin dephosphorylation.

Involucrin is a protein that forms a cornified envelope (CE), which is a solid structure that is insoluble in water and an indicator of skin-barrier protection. Involucrin is synthesized faster than other structural proteins and is produced from the stratum spinosum and stratum granulosum; it serves as a support for the crosslinking of other structural proteins [46]. Figure 4 indicates that the increase in thickness in the control group was not correlated with the expression of involucrin. However, ALA treatment resulted in an increase in involucrin expression, along with an increase in the thickness of the stratum corneum. ALA can assist in the differentiation and expression of KCs and enable the production of KC-forming proteins.

Loricrin is a glycine-serine-cysteine-rich protein synthesized from the stratum granulosum. When the differentiation of KCs is initiated, transglutaminase-1 binds to cytosolic proteins such as involucrin, cornifin, and loricrin inside the cell membrane to form a cornified envelope (CE), which performs the barrier function in the skin [47,48]. Although the IHC staining showed that loricrin appeared to be overstained as a whole, using the ImageJ program, we filtered only the DAB expression value in the region where the target protein was expressed on the basis of the range of high-expression intensity and compared the DAB-staining intensity values. The ALA treatment effect was the most pronounced in the day 3 group, and the groups in the other culture periods showed a relative decrease in loricrin expression when treated with ALA. This result confirmed that ALA did not significantly affect loricrin expression.

Keratin 10 belongs to a large series of intermediate-filament (IF) proteins that form the cytoskeleton with keratin 1 and major structural proteins of the epidermis [49]. Depending on the state of differentiation, KCs are represented by another set of keratins. In the stratum basale, keratinocytes express the K5/K14 pair; as the keratinocytes differentiate, they express the K1/K10 pair in the stratum spinosum. [50]. The IHC results for keratin 10 in Figure 6 showed the highest expression in HSEs treated with 10 μM ALA at day 7. It is inferred that this occurs because the stratum corneum becomes thicker, and the multiple layers constituting the stratum corneum are often differentiated. Because keratin 10 is expressed in the stratum granulosum of the epidermis, it is considered that ALA treatment affects the differentiation stage of the stratum corneum; thus, keratin 10 expression changes in ALA-treated HSEs. This phenomenon can be confirmed by the results for the ALA-treated HSEs on the 3rd day and 10 μM ALA-treated HSEs on the 7th day. We inferred that the keratin 10 expression rate was low in ALA-treated HSEs at day 5, because a part of the epidermis layer was shed owing to a problem with IHC staining, which was confirmed through histologically stained photographs. We attributed the lack of staining to the loss of a portion of the stratum corneum, on the basis of H&E staining and IHC results for other target proteins stained in the same sample block.

Collagen IV is a collagen type that is highly expressed at the interface between the epidermis and dermis and is present primarily in the basement membrane region of the skin. It is composed of alpha 1 (IV) and alpha 2 (IV) networks and is a macromolecule of the main skeleton of the basement membrane. Collagen IV molecules with such a complex network structure can play a role in scaffolding for cell adhesion and adhesion with glycoproteins in other matrices [51,52]. The 10 μM ALA-treated HSEs at day 5 and the ALA-treated group at day 7 exhibited a high expression of collagen IV, which indicated that ALA enabled the attachment of KCs, forming the epidermis layer and the attachment of the stratum corneum.

### 3.4. Effect of ALA Treatment on Protein Gene Expression in HSEs

ALA treatment generally resulted in increased expression of genes contributing to epidermal formation, including the COL1A gene, in the 1 μM ALA group at day 3 (Figure 8). However, under other experimental conditions, several other differences were observed, and the genes showing general expression showed similar expression to the IVL, KRT10, and COL1A genes. The expression of the IVL and KRT10 genes and the involucrin and KRT10 proteins (Figure 4, Figure 5, and Figure 8) indicate that the stratum corneum formation by KCs through skin recycling is the most active during the early stages of differentiation. It is presumed that the gene expression was high in order to synthesize the protein in the early stages of differentiation and that a relatively low gene expression was exhibited because differentiation was induced to a certain extent and sufficient protein was formed in the stratum corneum with the lapse of the culture period. This is because, as confirmed by H&E staining, the epidermis formation was similar between the 3rd day in the ALA-treated group and the 7th day in the control group. Significant changes are considered to occur in early expression while ALA affects the rapid differentiation of KCs and the early stage of stratum corneum formation.

## 4. Materials and Methods

### 4.1. Photolithography and Soft Lithography

Using photolithography technology, we fabricated an SU-8 master. SU-8 (MicroChem), which is a negative photoresist on a 4-inch Si wafer (Dasom RMS), was spin-coated to a thickness of 150 μm and subsequently baked. The photomask was aligned with a width of 200 μm and a height of 150 mm for which microfluidic channels were designed on the wafer and exposed to UV radiation so that the pattern designed for the photomask was formed on the wafer. Next, we baked and developed the pattern using the SU-8 developer to make the SU-8 master. Using the SU-8 master produced in this manner, a microfluidic channel pattern chip was produced via polydimethylsiloxane (PDMS) soft lithography technology, which is a method of molding and replicating PDMS SYLGARD 184 A and B at a ratio of 10:1 (Figure 9a). This microfluidic channel pattern was used in the lower PDMS chip layer in this study.

### 4.2. Fabrication of Pumpless Microfluidic Chip

The pumpless microfluidic skin-on-a-chip manufacturing process was described in a previous paper [31]. It consists of a slide glass, a lower PDMS chip layer, an upper PDMS chip layer, and a porous polyester membrane (0.4 μm pores, Corning Inc., Tewksbury, MA, USA) between the two chip layers (Figure 9a,b). Each configuration was O_2_ plasma bonded to a CUTE-1MP product (Femto Science Inc., Suwon, Korea), which is a plasma handler, and each layer was bonded strongly enough to seal between each floor via surface modification. Microfluidic channels patterned in the lower PDMS chip layers were designed to connect the storage of the upper chip. The upper PDMS chip was designed with a structure in which a cylindrical culture chamber with a diameter of 8 mm was located in the center of the chip and culture medium storage chambers were located on both sides and connected via a lower channel. The culture medium was designed to be perfused through the microfluidic channel of the lower chip and supplied to the 3D skin equivalent through the polyester membrane of the culture chamber.

### 4.3. Gravity Flow System

A gravity flow system was used to create a 3D culture in a fluid environment. As shown in Figure 9c, the system consists of a PC for driving, a motor, and a dish holder that can transmit movement to the chip, and a program is used to control the required tilt angle and rotation frequency. When the set time elapses, the badge can be effectively recirculated by causing the culture solution to flow in the opposite direction while tilting in the opposite direction. The flow rate of the culture medium in the microfluidic channel can be controlled mainly by adjusting the tilting angle, and the volume flow rate was 10 μL/min when the tilting angle was 15° [53].

### 4.4. Construction of a 3D Skin-Equivalent Model

As shown in Figure 10, rat-tail collagen type I (Corning), 10x DMEM, 0.5 N NaOH, primary human fibroblasts (FBs, Bio Solution Co Ltd.) suspension (final cell concentration 5.0 × 10^5^ cell/mL), and 1x DMEM were mixed to neutralize the gel. The rat-tail collagen concentration was fixed at 6.12 mg/mL. To fabricate the dermis layer, we seeded the collagen-FBs suspension on the chip to a height of 3 mm and deposited for 40 min at 37 °C in an incubator under 5% CO_2_. Thereafter, the dermal layer (DL) was cultured in DMEM (with 10% FBS and 1% penicillin/streptomycin) for 5 days, and the medium was changed every day. Next, primary human keratinocytes (KCs, Bio Solution Co Ltd., Seoul, Korea) suspension (final cell concentration 1.0 × 10^6^ cell/mL) were seeded on the dermis layer and cocultured for 2 days. KGM-Gold^TM^ Keratinocyte Growth Medium (Lonza) was supplied only above the DL-KCs, and DMEM was supplied to the channel of the chip. ALA supplies E-media that induces the differentiation of KCs for 3 to 7 days and at the same time provides an environment similar to that of real skin by exposing it to air (E-media composition: DMEM/Ham’s F12 (EGF-1 10 ng/mL, hydrocortisone 0.4 μg/mL, insulin 5 μg/mL, transferrin 5 μg/mL, 3,3,5-triiodo-L-thyonine sodium salt 2 × 10^−11^ M, cholera toxin 10^−10^ M, 10% (*v/v*) FBS, 1% penicillin/streptomycin). ALA treatment was performed at the AE stage (1 μM and 10 μM ALA). Human dermal FBs were used at passages 5–7, and human epidermal KCs were used at passages 4–6. All cultures were incubated at 37 °C in an incubator under 5% CO_2_. The medium was changed every day.

### 4.5. Measurement of Contraction of Skin Equivalents

We photographed the formation process of the 3D culture skin equivalents (SEs) seeded in the culture chamber of the pumpless microfluidic chip using a camera to measure the degree of contraction of the entire tissue area. The cultured tissue was observed every 24 h, and the area was measured using AutoCAD. Four samples were analyzed; the difference in contraction for each sample was determined by calculating the mean value, standard deviation, and *p* value for the values measured using the Prism software; and the data were visualized on a graph.

### 4.6. Hematoxylin and Eosin (H&E) and Immunohistochemistry (IHC) Staining

The samples were fixed in formaldehyde at 37 °C before being embedded in paraffin wax. The paraffin-embedded tissues were sectioned into 4-µm-thick sections using a microtome. Serial sections (4 μm) were mounted onto glass slides and subjected to either H&E staining or immunohistochemistry (IHC) analysis. For H&E staining, the sections were stained with hematoxylin solution (Sigma-Aldrich, St. Louis, MO, USA). After being washed with tap water, the sections were stained with eosin solution (Sigma-Aldrich Co.). The stained tissue was observed using an inverted optical microscope (Olympus, IX73-F22PH, Tokyo, Japan). For IHC, deparaffinized paraffin-embedded sections were incubated with primary antibodies specific to collagen IV (ab 53112), keratin 10 (ab76318), filaggrin (ab81468), loricrin (ab85679), and involucrin (ab53112) separately overnight at 4 °C; all antibodies were used at a 1:200 dilution and purchased from Abcam. We incubated the sections with horseradish peroxidase (HRP)-conjugated secondary antibody and visualized them using 3,3-diaminobenzidine (DAB) to detect the protein of interest. The slides were photographed using an optical microscope (Olympus, IX73-F22PH, Tokyo, Japan) attached to an optical camera (DP73-ST-SET; Olympus). Each sample was randomly analyzed five times, and IHC image analysis was performed using the free software ImageJ/Fiji. The measured values were analyzed using the Prism software.

### 4.7. RNA Extraction from Human Skin Equivalent

After adding 1 mL of TRIzol™ reagent (Invitrogen) to the cultured SE tissue, we homogenized the tissue, followed by the addition of 200 µL of chloroform and homogenization. The homogenates were centrifuged at 12,000× *g* for 15 min at 4 °C to obtain phase separation and transfer the upper layer (liquid phase) of the three separated layers to a fresh tube. The RNA was precipitated by mixing with isopropanol (0.5 mL) and incubating on ice for 10 min. Centrifuge for 10 min at 12,000× *g* at 4 °C. The supernatant was removed for RNA wash. The pellet was washed with 1 mL of 70% ethanol composed of RNase-free water (DEPC water). Centrifuge at 7500× *g* for 10 min at 4 °C. The supernatant was removed, and the RNA pellet was dissolved in an appropriate volume of RNase-free water. The quality and quantity of RNA were analyzed using a SpectraMax M2 microplate reader (Molecular Devices, LLC., San Jose, CA, USA).

### 4.8. Gene Quantitative Analysis Using Two-Step Real-Time PCR (qPCR)

Two-step q-PCR is initiated with the process of synthesizing mRNA into cDNA using RT-PCR. Using amfiRivert cDNA Synthesis Platinum Master Mix (GenDEPOT), RNA samples that had been appropriately analyzed for quantity and quality were synthesized into cDNA. qPCR was performed using the cDNA synthesized in the previous step and LightCycler^®^ 480 SYBR Green I Master (Roche) from LightCycler^®^ 480 Instrument II (Roche). Human 18s rRNA was used as a housekeeping gene. Values were calculated using the delta–delta CT method. The primers used for qPCR are listed in Table 1.

### 4.9. Analysis of mRNA Expression Using RT-PCR and Gel Electrophoresi

The synthesized cDNA amplified the desired target genes 18s rRNA, FLG, IVL, CK10, ITG1, and COL1A1 using AccuPower^®^ PCR PreMix & Master Mix (Bioneer). Table 1 presents the details of the base sequences of the primers used in this analysis. Gel electrophoresis was performed to confirm the PCR products. A 1.5% agarose gel was used for gel electrophoresis; the gel-staining solution was mixed with the loading sample and DNA ladder (Bioneer) and subjected to electrophoresis. The DNA ladder and DNA sample to be separated were loaded on a 1.5% agarose gel, and electrophoresis was performed at 100 V. The bands separated by electrophoresis were photographed using a gel doc system.

### 4.10. Statistical Analysis

Data are expressed as the mean ± standard deviation (SD). Statistical analysis was performed using Prism. Statistical significance was determined using a two-way ANOVA. A *p* value < 0.05 was considered statistically significant. Different significance levels (*p* values) are indicated by asterisks.

## 5. Conclusions

In order to prevent aging, it is important to have an antioxidant action that can prevent the oxidative action that occurs in tissues. There are five major antioxidants that have a major effect on our body: α-lipoic acid, glutathione, vitamin C, vitamin E, and CoQ10. Among them, α-lipoic acid gives one electron to remove the active ingredient that promotes the aging action of the body and oxidizes itself. The component that interacts with it then gives the electron again. It prevents aging and fills the deficient antioxidant component so that the structure that oxidizes itself continues. HSEs were cultured using a pumpless-skin-on-a-chip, and α-lipoic acid was treated by concentration to test its effectiveness. The contraction rate of α-lipoic acid by concentration appeared differently, and the shrinkage rate was particularly fast when treated with a high concentration of 10 uM. It is inferred that this is due to continuous paracrine signaling with KC-FB, which promotes cell-mediated contraction and increases the rate of contraction. It is considered that α-lipoic acid contributes to the division and differentiation of KCs and is involved in the mechanism of forming epidermis. The result is that α-lipoic acid, as an antioxidant, aids in skin composition, protects the skin such as by forming a barrier to the development of the stratum corneum, and regulates water, nutrients, and external bacteria. It is believed that it can be given to help prevent the penetration of factors such as viruses. Changes in the proteins of Filaggrin, involucrin, loricrin, keratin10, and collagen IV by α-lipoic acid treatment were confirmed. Treatment of α-lipoic acid for as long as 7 days showed an increase in filaggrin, involucrin, and collagen IV proteins, and it was confirmed that the expression of keratin10 protein increased at some concentrations. Based on these results, a simple and convenient drivable pumpless skin-on-a-chip technology can be used to evaluate the efficacy of the drug and identify physiological changes as a skin substitute. We anticipate that this system can be used as an alternative to disease models and animal testing methods that are useful in the process of cosmetic development.

## Figures and Tables

**Figure 1 ijms-22-02160-f001:**
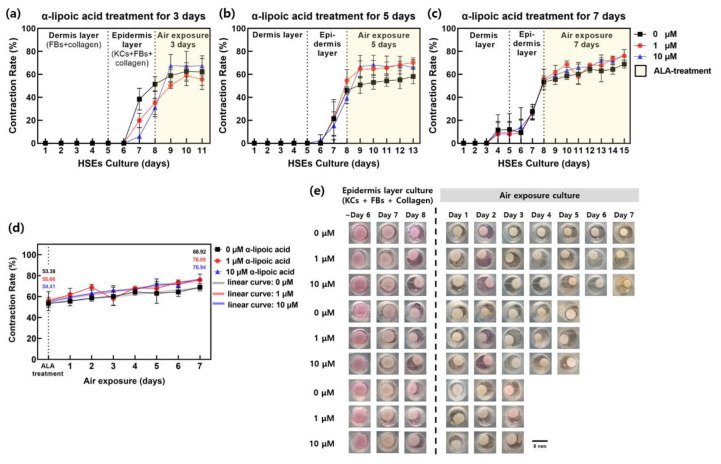
Variation of the contraction rate of the samples as a function of the air exposure 3, 5, and 7 days with the different concentrations (0, 1, 10 µM) of α-lipoic acid. The graph is the contraction rate of the group treated with α-lipoic acid for (**a**) 3 days, (**b**) 5 days, and (**c**) 7 days under air exposure culture conditions. The legends in the graphs (**a**–**c**) are the same. (■, 0 µM α-lipoic acid; ●, 1 µM α-lipoic acid; and ▲, 10 µM α-lipoic acid) (**d**) It is a graph comparing only the contraction rate of the 8th day, which is the period of coculture to dermis and epidermis, and the air exposure period of which α-lipoic acid was treated (maximum processing period for α-lipoic acid is 7 days). The change was represented by a linear curve during the α-lipoic acid processing period. (**e**) Photographic images of the samples following air exposure for the different concentrations of α-lipoic acid (bar = 8 mm).

**Figure 2 ijms-22-02160-f002:**
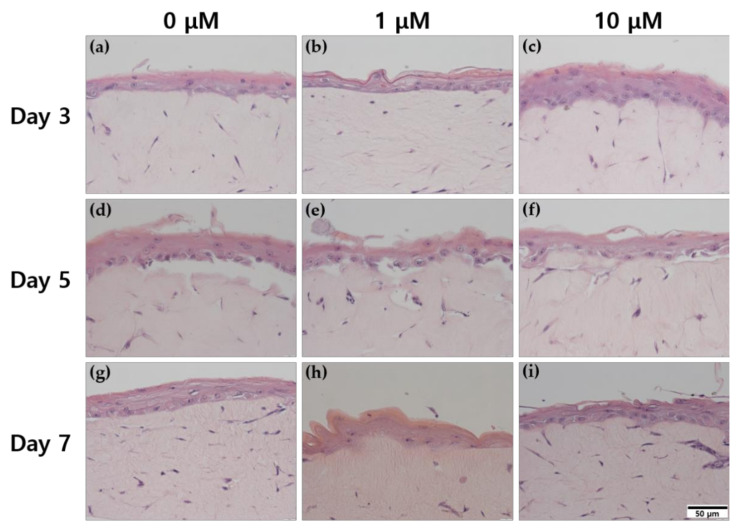
Hematoxylin and eosin staining (H&E) images of human skin equivalents (HSEs) for 3 days air exposure culture treated α-lipoic acid concentration of (**a**) 0 µM, (**b**) 1 µM, and (**c**) 10 µM. For 5 days, air exposure culture treated α-lipoic acid concentration of (**d**) 0 µM, (**e**) 1 µM, and (**f**) 10 µM. Seven days air exposure culture treated α-lipoic acid concentration of (**g**) 0 µM, (**h**) 1 µM, and (**i**) 10 µM. The results were taken at 40 magnification (Scale bar = 50 µm).

**Figure 3 ijms-22-02160-f003:**
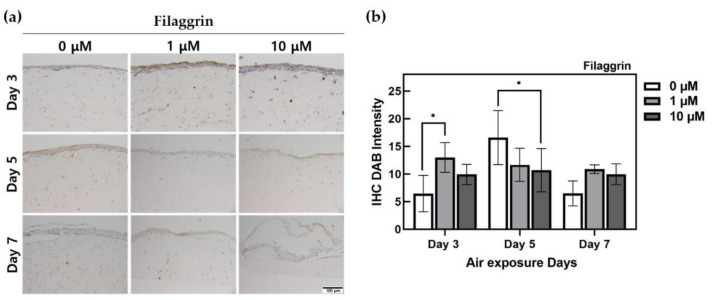
Immunohistochemistry-stained images for filaggrin of HSEs treated α-lipoic acid concentration of 0, 1, and 10 μM. (**a**) This is the result of filaggrin expressed by IHC staining in HSEs. The x-axis is listed by ALA-treated concentration, and the y-axis is listed by ALA-treated period. The scale bar is 100 μm. (**b**) This is a graph of quantification of 3,3′-Diaminobenzidine intensity for IHC staining of filaggrin protein using the ImageJ program. (*n* = 4; *, *p* < 0.05 vs. 0 μM for each period).

**Figure 4 ijms-22-02160-f004:**
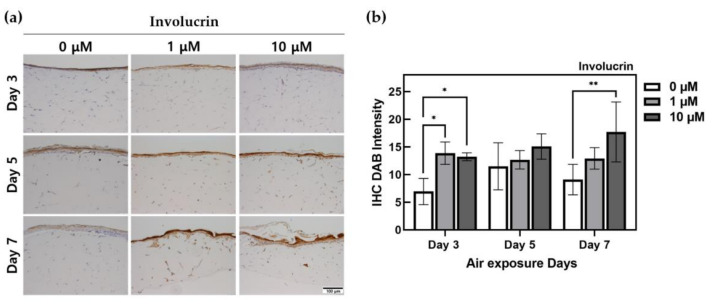
Immunohistochemistry-stained images for involucrin of HSEs treated α-lipoic acid concentration of 0, 1, and 10 μM. (**a**) This is the result of involucrin expressed by IHC staining in HSEs. The x-axis is listed by ALA-treated concentration, and the y-axis is listed by ALA-treated period. The scale bar is 100 μm. (**b**) This is a graph of quantification of DAB intensity for immunohistochemistry staining of involucrin protein using the ImageJ program. (*n* = 4; *, *p* < 0.05; **, *p* < 0.01 vs. 0 μM for each period).

**Figure 5 ijms-22-02160-f005:**
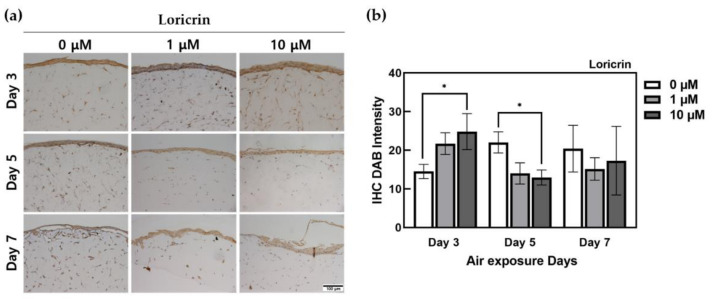
Immunohistochemistry-stained images for loricrin of HSEs treated α-lipoic acid concentration of 0, 1, and 10 μM. (**a**) This is the result of loricrin expressed by IHC staining in HSEs. The x-axis is listed by ALA-treated concentration, and the y-axis is listed by ALA-treated period. The scale bar is 100 μm. (**b**) This is a graph of quantification of DAB intensity for immunohistochemistry staining of loricrin protein using the ImageJ program. (*n* = 4; *, *p* < 0.05 vs. 0 μM for each period).

**Figure 6 ijms-22-02160-f006:**
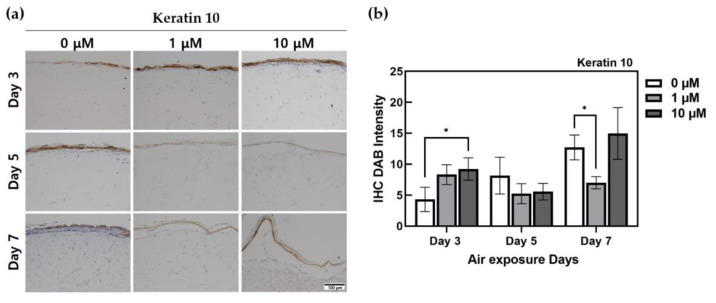
Immunohistochemistry-stained images for keratin 10 of HSEs treated α-lipoic acid concentration of 0, 1, and 10 μM. (**a**) This is the result of keratin 10 expressed by IHC staining in HSEs. The x-axis is listed by ALA-treated concentration, and the y-axis is listed by ALA-treated period. The scale bar is 100 μm. (**b**) This is a graph of quantification of DAB intensity for immunohistochemistry staining of Keratin 10 protein using the ImageJ program. (*n* = 4; *, *p* < 0.05 vs. 0 μM for each period).

**Figure 7 ijms-22-02160-f007:**
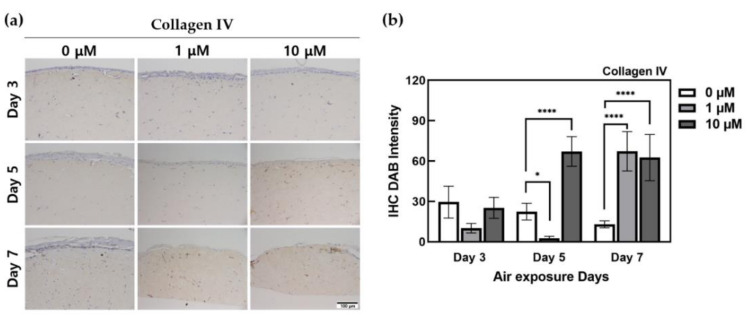
Immunohistochemistry-stained images for collagen IV of HSEs treated α-lipoic acid concentration of 0, 1, and 10 μM. (**a**) This is the result of collagen IV expressed by IHC staining in HSEs. The x-axis is listed by ALA-treated concentration, and the y-axis is listed by ALA-treated period. The scale bar is 100 μm. (**b**) This is a graph of quantification of DAB intensity for immunohistochemistry staining of collagen IV protein using the ImageJ program. (*n* = 4; *, *p* < 0.05; ****, *p* < 0.0001 vs. 0 μM for each period).

**Figure 8 ijms-22-02160-f008:**
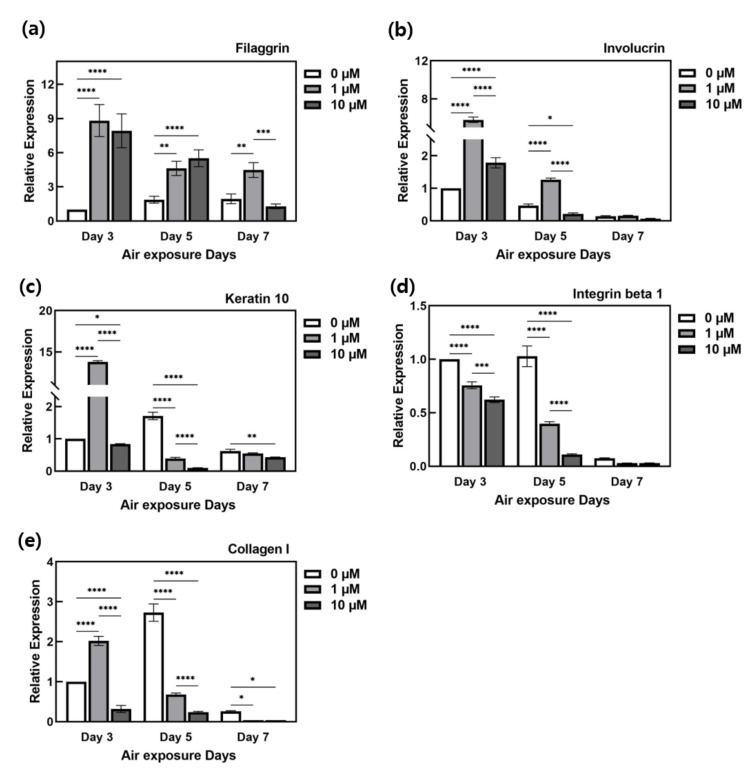
Relative gene expression of (**a**) filaggrin, (**b**) involucrin, (**c**) keratin 10, (**d**) integrin beta 1, and (**e**) collagen I with varing α-lipoic acid treated concentration and period by real-time q-PCR. It was quantified by relative comparison based on HSEs treated with 0 μM α-lipoic acid for 3 days. (*n* = 3; *, *p* < 0.05; **, *p* < 0.01; ***, *p* < 0.001; ****, *p* < 0.0001).

**Figure 9 ijms-22-02160-f009:**
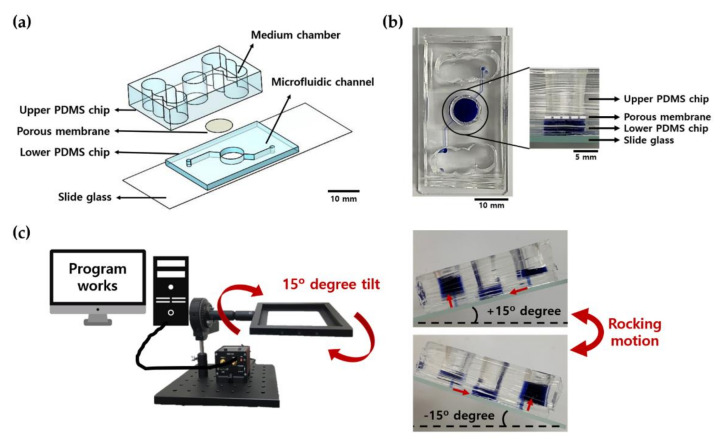
Schematic diagram of the pumpless skin-on-a-chip (SOC) and gravity flow system in action. (**a**) Configuration diagram of pumpless microfluidic skin-on-a-chip. (**b**) Actual appearance of pumpless SOC. The microfluidic channel of the lower chip is filled with blue dye and an enlarged side view of the culture chamber. (**c**) Schematic diagram of the operation of the gravity flow system. It works by shaking both sides at 15°. The 15° slope allows the medium to circulate through the microfluidic channels.

**Figure 10 ijms-22-02160-f010:**
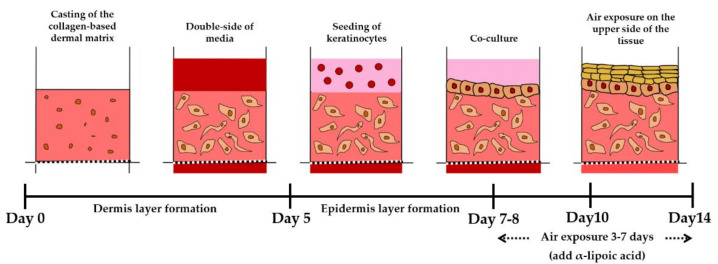
Schematic diagram of the 3D skin-model formation process in pumpless skin-on-a-chip (SOC). Skin equivalents are formed through air exposure on the 7th to 8th days after the initial culture is started, and at this time, a drug test is conducted by treating the culture medium with α-lipoic acid.

**Table 1 ijms-22-02160-t001:** Sequences of forward and reverse primers used for qPCR analysis of gene expression levels.

Gene	Forward Primer	Reverse Primer
18s rRNA	5′-GGCGCCCCCTCGATGCTCTTAG-3′	5′-GCTCGGGCCTGCTTTGAACACTCT-3′
Filaggrin	5′-GGAGTCACGTGGCAGTCCTCACA-3′	5′-GGTGTCTAAACCCGGATTCACC-3′
Involucrin	5′-CCGCAAATGAAACAGCCAACTCC-3′	5′-GGATTCCTCATGCTGTTCCCAG-3′
Keratin 10	5′-CCGGAGATGGTGGCCTTCTCTCT-3′	5′-GGCCTGATGTGAGTTGCCATGCT-3′
Intergrin beta 1	5′-CAAGAGAGCTGAAGACTATCCCA-3′	5′-TGAAGTCCGAAGTAATCCTCCT-3′
Collagen 1a1	5′-CCAGCGCTGGTTTCGACTTCA-3′	5′-CCATGTTCTCGATCTGCTGGCT-3′

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
