# Peer review of "Effect of α-Lipoic Acid on the Development of Human Skin Equivalents Using a Pumpless Skin-on-a-Chip Model"

_ijms, 2021, doi:10.3390/ijms22042160_

Round 1
Reviewer 1 Report
In this manuscript the authors presented a very interesting methods to evaluate the efficacy of cosmetics using skin equivalent, and some comments have arised from that study.
1) In Fig 1, the authors are evaluating the % of contraction. It would be great to explain as introduction of the corresponding result paragraph why is it important to evaluate that parameter.
2) In the Fig 2, the authors mentioned that ALA treatment induced difference in FBs activity. Which activity are the authors speaking about? Please precise it in the text.
3) "When treated for 7 days, the number of cells forming the dermis and epidermis at 10 μM was relatively large." What the authors are meaning by large? Higher number of cells? If yes, it does not seems to be the case between the control and 10uM conditions (the number of nuclei seems to be lower at day 7 than day 3 for the 10uM). Do the authors have performed any cell counting to support this claim?
4)How the authors explained that Filaggrin expression is significantly higher in the control than in the 10uM condition for 5 days? Also, any explanation for the high expression level of Filaggrin at day 5 compared to day 3 and 7 for the vehicle condition?
5) How the authors explain the reduced expression of Loricrin with the time of culture for ALA treatment while it is progressively increased for the control condition? Is this reflecting a defect in epidermal differentiation with ALA treatment?
6) In general, for the different parameters evaluated in IHC, a decreased is observed at day 5 compared to day 3 and 7 mainly for the 1uM condition. Do the authors have any explanation? Did any problem appeared during the culture (e.g.increased apoptosis)?
7) it would have been interesting to have the IHC and qPCR data within the same figure to compare both expression levels.
8) "Histological changes with respect to antioxidants were confirmed using H&E staining with ALA treatments at 1 and 10 μM (Figure 2)". Did the authors evaluated expression of antioxidants? If yes, it would be great to present the data to justify this claim.
9) "In the control group, which 390 was treated with 0 μM ALA, the expression level of filaggrin decreased depending on the culture period". Based on the data presented, this claim in wrong. The authors observed an increased expression at day 5 but the expression levels at day 3 and 7 are identical. Please revised the sentence accordingly.
10) "We attributed the lack of staining to the loss of a portion of the stratum corneum". How the loss of stratum corneum can explain the lack of staining in the stratum granulosum where K10 is expressed?
11) It would be great to add the reference areas on the representative pictures for the difference IHC stainings to understand where exactly the intensity measurement have been performed.
Minor comments:
P3, l130: "pumpless SOC was used skin equivalents". I think that was should be deleted.
P4, l147:(b) 5 dyas. Please correct by days.
p7, l255: significant difference (p < 0.05): the star is missing.
Author Response
Reply to Reviewer 1
We appreciate the thorough review of our manuscript. We revised the manuscript extensively for clarity according to the Reviewer's comments.
Comments and Suggestions for Authors from Reviewer 1
In this manuscript the authors presented a very interesting methods to evaluate the efficacy of cosmetics using skin equivalent, and some comments have arised from that study.
1) In Fig 1, the authors are evaluating the % of contraction. It would be great to explain as introduction of the corresponding result paragraph why is it important to evaluate that parameter.
Answer : We added why contraction is important in line with the comments at the first paragraph of 2.1. as follows. “Cell-mediated collagen contraction occurs in 3D culture, which is known to be similar to the process that occurs when cells interact with protein fibers and collagen in 3D culture[36]. It seems clear that the contraction phenomenon appears depending on the activity of fibroblasts[36]. Paracrine factors secreted by dermal fibroblasts fine-tune the balance between keratinocyte proliferation, migration, and differentiation via dual paracrine signals when ball-cultured with keratinocytes[37]. It appears to affect contractions such as ECM synthesis and dermal remodeling[38]. The contraction of the collagen scaffold can be used as an assay to determine whether the co-cultured tissue is properly cultured. Shrinkage rate is measured as an indicator that can monitor a real-time sample of fibroblasts-collagen (the reason for the experiment, no contraction when cells die) corresponding to the dermis layer of cultured tissue with no cell activity or reproducibility.”
2) In the Fig 2, the authors mentioned that ALA treatment induced difference in FBs activity. Which activity are the authors speaking about? Please precise it in the text.
Answer : We added the contents according to the comment. “there was differences in the number and shape of FBs cells were observed in the dermis layer depending on the presence or absence of α-lipoic acid. Many elongated cells are observed that can be inferred to have high activity of most FBs in the ALA-treated group, and more fiberous morphology is observed around the FBs.”
3) "When treated for 7 days, the number of cells forming the dermis and epidermis at 10 μM was relatively large." What the authors are meaning by large? Higher number of cells? If yes, it does not seems to be the case between the control and 10uM conditions (the number of nuclei seems to be lower at day 7 than day 3 for the 10uM). Do the authors have performed any cell counting to support this claim?
Answer : In the case of the day 3, the difference in the number of FBs cells was not significantly displayed between the tissues treated with 0 uM and 10 uM ALA, which were controlled by the dermis layer. The Epidermis layer revealed a clear increase in thickness and an increase in the number and differentiation of KCs cells. This content has already been mentioned in the text. On the day 7, it was possible to see a large increase in the number of FBs in HSEs treated with 10 uM ALA compared to the control. Among the total HSEs, the highest number of FBs at 10uM on the day 7 was revealed. In the case of KCs in the Epidermis layer, a larger number of cells were observed at 10 uM than the control. This content is explained in this way in the text.
“When treated for 7 days, the number of cells forming the dermis and epidermis at 10 μM was relatively large. Regarding the changes in tissue morphogenesis according to the culture period in the control group, a longer culture period resulted in a more prominent epidermis layer formation and an increase in number and activity of FBs in the dermis layer.“
4) How the authors explained that Filaggrin expression is significantly higher in the control than in the 10uM condition for 5 days? Also, any explanation for the high expression level of Filaggrin at day 5 compared to day 3 and 7 for the vehicle condition?
Answer : The results are shown in the graph of Fig. 3, and the expression of filaggrin was measured higher by control than 10 uM from the day 5. The exact reason or cause of filaggrin being specified higher than other days on Day 5 is not clear. We are reporting the results that came out when processing ALA on our platform.
5) How the authors explain the reduced expression of Loricrin with the time of culture for ALA treatment while it is progressively increased for the control condition? Is this reflecting a defect in epidermal differentiation with ALA treatment?
Answer : As a result, on the day 5 and day 7, the expression of loricrin was relatively low in the ALA-treated group compared with 0 uM of control. In the Control group, we can see that after 3 days there is an increase in loricrin. It cannot be asserted that this reflects the defect of epidermal differentiation due to ALA treatment. In the case of Involucrin and filaggrin, the ALA-treated group showed higher expression. If there is a defect in epidermal differentiation, protein expression should also be low. I think it simply means that the expression of loricrin is low.
6) In general, for the different parameters evaluated in IHC, a decreased is observed at day 5 compared to day 3 and 7 mainly for the 1uM condition. Do the authors have any explanation? Did any problem appeared during the culture (e.g.increased apoptosis)?
Answer : From the IHC results, it is correct that overall low expression appeared at 1 uM condition on day 5. There were no problems during culturing. If there were problems during incubation, such as increased self-destruction of cells, the contraction rate of HSE would have shown a low contraction. There were no problems in the process of culturing HSEs and treating them with ALA. This result is obtained from 4 more HSEs samples.
7) it would have been interesting to have the IHC and qPCR data within the same figure to compare both expression levels.
Answer : Thank you for your good comments. The reason I didn't compare both data together in the text is because We thought the gene expression couldn't be seen as identical to the protein expression. Too many factors are involved in seeing mRNA as equivalent to protein expression, so I thought it would be very restrictive to build and interpret both levels associating them. Gene expression can predict protein expression in the future, but I don't think it can be claimed to be actual protein expression. A related treatise was reported. It seems that understanding is quick if you refer to this paper. Schwanhäusser, B.; Busse, D.; Li, N.; Dittmar, G.; Schuchhardt, J.; Wolf, J.; Chen, W.; Selbach, M. Global quantification of mammalian gene expression control. Nature 2011, 473, 337–342. http://www.nature.com/nature/journal/v473/n7347/full/nature10098.html
In this paper, found that mRNA levels account for about 40% of protein level variability. Changes in the gene level are potentially significant because changes in these HSEs are expected. We think the expression of the protein is meaningful because it can determine the level of the protein constituting in the actual HSEs.
8) "Histological changes with respect to antioxidants were confirmed using H&E staining with ALA treatments at 1 and 10 μM (Figure 2)". Did the authors evaluated expression of antioxidants? If yes, it would be great to present the data to justify this claim.
Answer : We think that mentioning "confirmed" is an incorrect expression. we revised it to the inferred content with reference to the content you commented. “From the results of H & E in Fig. 2, in the case of the ALA-treated group compared to the control, a relatively clear differentiation was seen in the epidermis, and it was found that the dermis was also formed into an elongated shape showing activity, which could better show the network between FBs. These results can be inferred that the addition of ALA served as an antioxidant.”
9) "In the control group, which 390 was treated with 0 μM ALA, the expression level of filaggrin decreased depending on the culture period". Based on the data presented, this claim in wrong. The authors observed an increased expression at day 5 but the expression levels at day 3 and 7 are identical. Please revised the sentence accordingly.
Answer : We admit that there is an error in the content described in the text. This part will be deleted. Thank you for the good point out.
10) "We attributed the lack of staining to the loss of a portion of the stratum corneum". How the loss of stratum corneum can explain the lack of staining in the stratum granulosum where K10 is expressed?
Answer : In the process of IHC staining, a part of the epidermis layer of 1 and 10 uM on the day 5 was damaged. This can be confirmed on the photograph through the hematoxylin-stained part that stains the core part of the cell. Day 5 1, 10 uM epidermis layer lost the part where the core of KCs should be. It can be seen that the lower part of the Corneum layer was accidentally lost due to staining. As shown in Fig.2 ~ 5 and 7, the results of H & E and filaggrin, involucrin, loricirin and collagen IV, there was no problem in HSEs production and paraffin block production because all the epidermis layer appeared to be stained. It seems that K10 disappeared during the staining reagent treatment process during IHC staining.
11) It would be great to add the reference areas on the representative pictures for the difference IHC stainings to understand where exactly the intensity measurement have been performed.
Answer : In the case of IHC staining, the expression of the target protein must be compared considering only the DAB staining intensity and area of the target. For a valid comparison, the strength and area of the non-target area was removed using the Image J program. The brown color corresponding to DAB of the Image J program was filtered to quantify the degree of DAB staining. Therefore, the only valid area is the brown area. Only that part was analyzed.
Minor comments:
P3, l130: "pumpless SOC was used skin equivalents". I think that was should be deleted.
P4, l147:(b) 5 dyas. Please correct by days.
p7, l255: significant difference (p < 0.05): the star is missing.
Answer : We corrected all. Thanks for minor comments
Reviewer 2 Report
Manuscript ID: ijms-1084148
Anti-aging efficacy of α-lipoic acid in skin model 2 using a pumpless skin-on-a-chip
The manuscript reports a skin-on-chip pumpless system capable of testing the in vitro efficacy of α-lipoic acid in a skin model. I think the topic and results are interesting for the scientific community. Therefore, I recommend publication after a few minor suggestions and corrections.
The next comments are thoughts that I had when I read the manuscript and it would be nice to add them, in case the others reader have the same thoughts.
- How long it takes for the fluid to pass from one chamber to the other?
- Does air get in and blocks the flow? How did you solve this problem?
- How often you introduce new medium and how?
Corrections
- In figure 1 capture you can read “5 dyas” instead of days.
- Is there any reason why there isn’t a conclusion and the materials and methods are in the end? If think it will help the reader to start with materials and methods, results, discussion and at last conclusion.
Author Response
Reply to Reviewer 2
We appreciate the thorough review of our manuscript. We revised the manuscript extensively for clarity according to the Reviewer's comments.
Comments and Suggestions for Authors from Reviewer 2
Manuscript ID: ijms-1084148
Anti-aging efficacy of α-lipoic acid in skin model 2 using a pumpless skin-on-a-chip
The manuscript reports a skin-on-chip pumpless system capable of testing the in vitro efficacy of α-lipoic acid in a skin model. I think the topic and results are interesting for the scientific community. Therefore, I recommend publication after a few minor suggestions and corrections.
The next comments are thoughts that I had when I read the manuscript and it would be nice to add them, in case the others reader have the same thoughts.
- How long it takes for the fluid to pass from one chamber to the other?
Answer : The papers referred to in the text "[53] Lee, S .; Jin, S.-P .; Kim, YK; Sung, GY; Chung, JH; Sung, JH Construction of 3D multicellular microfluidic chip for an in vitro skin model. Biomed. Microdevices 2017 , 19, 22" already mentioned the flow velocity of this chip. Tilting angle of 15 ° was applied to achieve the flow rate of 10 μl/min with this pumpless chip. It was executed in the same way as this reference paper. Therefore, in this study as well, the fluid was supplied according to this flow velocity. The operation of the gravity flow system takes 2 minutes for one set that forms a tilt of 15 degrees, then forms 15 degrees, and maintains parallelism again.
- Does air get in and blocks the flow? How did you solve this problem?
Answer : No air flows into the channels during culture. Before starting the 3D culture on the chip, the lower microfluidic channel is filled with the culture solution first, and at this time, only the culture solution is filled so that air bubbles do not occur. In this way, if you proceed with 3D culture with the culture solution filled first, the air in the culture will not flow in. If air enters the channel at this time, use a pipette to flow a new culture solution toward the inlet side and pull out the air bubble to the outlet. It can be easily removed.
- How often you introduce new medium and how?
Answer : The new culture medium was added once every 1-2 days on average. During culture, if cell division is active or the amount of cells put in is large, fresh culture medium should be added 1-2 times a day. The badges are rapidly oxidized by cell waste, so we add new cultures more often. (Since it is essential to supply a fresh culture solution in the culture, it needs to be adjusted according to the culture condition.) We added answers to comments to material and methods.
Corrections
- In figure 1 capture you can read “5 dyas” instead of days.
Answer : There was a typo. It's corrected.
- Is there any reason why there isn’t a conclusion and the materials and methods are in the end? If think it will help the reader to start with materials and methods, results, discussion and at last conclusion.
Answer : We added conclusion based on feedback. The reason for ending with materials and methods is that they were created in the order presented in the book, INTERNATIONAL JOURNAL OF MOLECULAR SCIENCES (IJMS) template.
Reviewer 3 Report
The authors have presented a thorough examination of the effects of a-lipoic acid on their human in vitro skin model using a pumpless skin-on-chip system. Functional changes (contraction) as well as protein and gene expression changes were tracked throughout the study to demonstrate the strength of the platform to study changes over time with/without treatment. However, the total effect of a-lipoic and air exposure seemed to differing degress of influence depending on which metric was used, which may weaken the robustness of this model. This manuscript deserved publication in this journal after addressing the following comments:
Overall
- Given the effect of alipoic acid that was strongest in the 3 day group for many of the metrics, I suggest focusing the paper on the strength of the platform as a means to be able to test things and focus less on justifying that alipoic acid has an effect.
- The title isn't reflective of the work demonstrated and may be misleading. Since the experiments focused on the effect of alipoic acid on skin development in vitro, any claims of "anti-aging" should be removed, as these were not directly measured in the study. I would recommend the title be changed to something like "a-lipoic acid [modulates/enhances/accelerates/influences] the [formation/development/maturation] of human skin equivalents using a pumpless skin-on-a-chip model"
Abstract
- replace "genotyping" with "gene expression"...genotyping refers to differences in the DNA (e.g., mutations, snps, etc), whereas gene epxression refers to mRNA expression which is what was done here
Methods
- Please include the source of the cell types that were used in the study
Results
- For Figure 1, please add representative pictures that were used to measure the contraction % (similar to fig2 of the author's coenzyme Q10 paper). It is not immediately clear as to what kind of contraction this represents so this would help show that. Also please indicate on the graphs when the a-lipoic acid was added in the timeline to make this more clear.
- For contraction rate, there seems to be a notable variation in the 0um dose between treatment groups and the overall effect of any dose of alipoic acid. Why does the 7day AE group have contraction changes in the early days? If i have understood correctly, day0 to day11 should be the same between all treatment groups and all conditions. If there are differences, that suggests overall variablity of the platform or artifacts of certain groups, which could be helpful when understanding what a "good" experiment looks like when using this platform. For example, if contraction happens within the first week, maybe those samples should not be used for testing or of treatment groups are at different contraction % at the time of treatment, that should be considered a covariate when calculating total effect of treatment (e.g., higher contraction at start of 0 dose treatment in day 3 group may be why it didnt change as much as 10um group). Please comment on this in the manuscript.
- I would also suggest adding another graph to this figure that plots the change from day 8 (i.e., normalized to day8) so that the effect of alipoic acid can be seen, which would better visualize what is discussed in the last paragraph of section 2.1
- FIgure 8D- integrin is mispelled as "intergrin". Please correct to read "Integrin B1"
- It would've been interesting to see if ALA increased the expression of γ-glutamyl cysteine ligase (γ-GCL), as mentioned in the introduction. THis would help to support the direct effect of this antoxidant's in this model
Discussion
- for gene expression, it is worth noting that gene expression decreasing with time may also reflect a change in the cell number or cell ratio that express the measure gene, thereby change the overall expression (since expression is normalized to housekeeping genes that are expressed in all cells)
Author Response
Reply to Reviewer 3
We appreciate the thorough review of our manuscript. We revised the manuscript extensively for clarity according to the Reviewer's comments.
Comments and Suggestions for Authors from Reviewer 3
The authors have presented a thorough examination of the effects of a-lipoic acid on their human in vitro skin model using a pumpless skin-on-chip system. Functional changes (contraction) as well as protein and gene expression changes were tracked throughout the study to demonstrate the strength of the platform to study changes over time with/without treatment. However, the total effect of a-lipoic and air exposure seemed to differing degress of influence depending on which metric was used, which may weaken the robustness of this model. This manuscript deserved publication in this journal after addressing the following comments:
Overall
- Given the effect of alipoic acid that was strongest in the 3 day group for many of the metrics, I suggest focusing the paper on the strength of the platform as a means to be able to test things and focus less on justifying that alipoic acid has an effect.
Answer : We reduced the content of ALA effects and mentioned the strengths of the platform in 106 ~ 111
- The title isn't reflective of the work demonstrated and may be misleading. Since the experiments focused on the effect of alipoic acid on skin development in vitro, any claims of "anti-aging" should be removed, as these were not directly measured in the study. I would recommend the title be changed to something like "a-lipoic acid [modulates/enhances/accelerates/influences] the [formation/development/maturation] of human skin equivalents using a pumpless skin-on-a-chip model"
Answer : We change the title of the paper into “Effect of a-lipoic acid on the development of human skin equivalents using a pumpless skin-on-a-chip model”. In line with the comments, we reduced the content of ALA effects and mentioned the strengths of the platform.
Abstract
- replace "genotyping" with "gene expression"...genotyping refers to differences in the DNA (e.g., mutations, snps, etc), whereas gene epxression refers to mRNA expression which is what was done here
Answer : We changed it according to the comment.
Methods
- Please include the source of the cell types that were used in the study
Answer : Human-derived primary cells were used. We also added this content to the text.
Results
- For Figure 1, please add representative pictures that were used to measure the contraction % (similar to fig2 of the author's coenzyme Q10 paper). It is not immediately clear as to what kind of contraction this represents so this would help show that. Also please indicate on the graphs when the a-lipoic acid was added in the timeline to make this more clear.
Answer : To help you understand where the contraction was measured, we have added photos of the actual HSEs measured by period to Figure 1e. And the time when ALA was added is displayed in yellow box in Figures 1, a, b and c.
- For contraction rate, there seems to be a notable variation in the 0um dose between treatment groups and the overall effect of any dose of alipoic acid. Why does the 7day AE group have contraction changes in the early days? If i have understood correctly, day0 to day11 should be the same between all treatment groups and all conditions. If there are differences, that suggests overall variablity of the platform or artifacts of certain groups, which could be helpful when understanding what a "good" experiment looks like when using this platform. For example, if contraction happens within the first week, maybe those samples should not be used for testing or of treatment groups are at different contraction % at the time of treatment, that should be considered a covariate when calculating total effect of treatment (e.g., higher contraction at start of 0 dose treatment in day 3 group may be why it didnt change as much as 10um group). Please comment on this in the manuscript.
Answer : Some differences will be displayed depending on the condition of the cell. I don't think it's a big problem to see that the continuous contraction does not occur after the initial contraction, but is retained after the initial contraction. The initial contraction is meaningless because it is at a smaller level than the final contraction. I think the thing to consider in terms of shrinkage is a "bad" experiment. Experiments with extremely low shrinkage are the ones that need to be excluded. This is because cell death and cell partitioning become so bad that contraction does not occur. Through previous research, we have confirmed that such a pattern is displayed.
- I would also suggest adding another graph to this figure that plots the change from day 8 (i.e., normalized to day8) so that the effect of alipoic acid can be seen, which would better visualize what is discussed in the last paragraph of section 2.1
Answer : A representative graph of changes in contraction during the ALA treatment period has been added to make it easier to explain the results of 2.1 (figure1d). If you include the degree of change for 7 days, it can be explained in the sense that it includes all the data for 3 to 7 days. The degree of change is shown in the linear curve.
- FIgure 8D- integrin is mispelled as "intergrin". Please correct to read "Integrin B1"
Answer : We changed it according to the comment.
- It would've been interesting to see if ALA increased the expression of γ-glutamyl cysteine ligase (γ-GCL), as mentioned in the introduction. THis would help to support the direct effect of this antoxidant's in this model
Answer : Thank you for your helpful comment.
Discussion
- for gene expression, it is worth noting that gene expression decreasing with time may also reflect a change in the cell number or cell ratio that express the measure gene, thereby change the overall expression (since expression is normalized to housekeeping genes that are expressed in all cells)
Answer : We think it can be ruled out to some extent that it reflects changes in the number or proportion of cells. This is because after extracting the mRNA, the same dilution was made based on the amount measured using nanodrop, the concentration was adjusted, and then the mRNA values of all the samples were used the same so as to synthesize the cDNA. All conditional comparisons were made overall in the same environment, as the same amount of mRNA was normalized using the housekeeping gene to the combine.
Round 2
Reviewer 1 Report
The authors answered most of my comments however I have a few comments which still need to be answered:
1) "When treated for 7 days, the number of cells forming the dermis at 10 μM was relatively large." Please indicate if the number was increased compared to Day 3 and or 5 or is this result referring to differences with 0 and/or 1uM. Also please indicate on what that observation is based, did the authors counted the number of nuclei?
2)"These results can be inferred that the addition of ALA served as an antioxidant.” How the authors can claim that ALA served as antioxidant? Did the authors have measured ROS levels?
Author Response
We appreciate the thorough review of our manuscript twice. We revised the manuscript according to the your comments.
Comments and Suggestions for Authors
The authors answered most of my comments however I have a few comments which still need to be answered:
1) "When treated for 7 days, the number of cells forming the dermis at 10 μM was relatively large." Please indicate if the number was increased compared to Day 3 and or 5 or is this result referring to differences with 0 and/or 1uM. Also please indicate on what that observation is based, did the authors counted the number of nuclei?
Answer : Tried to explain that there was a relative increase at 10 uM compared to 0 and / or 1 uM. To make the explanation concrete, I modified the text as follows. “When treated for 7 days, the number of cells forming the dermis at 10 μM was relatively larger than at 1 μM and/or 0 μM.”
The cell core of the dermis layer was manually counted using the cell counter plugin of the Image J program. The parts with the cell shape were counted, excluding the parts where the proper nucleus shape was not visible. Magnification was counted based on 20x captured images. (Because the 40x magnification is partially enlarged, the image of the whole organization is not shown.) 7 days 0 uM was 38, 1 uM was 38, and 10 uM was 50. (Number of fibroblast cells in a 20X image of skin equivalents) Added content to the text. “7 days 0 uM was 38, 1 uM was 38, and 10 uM was 50. (Number of fibroblast cells in a 20X image of skin equivalents)”
2)"These results can be inferred that the addition of ALA served as an antioxidant.” How the authors can claim that ALA served as antioxidant? Did the authors have measured ROS levels?
Answer : ROS level was not measured. We agree the claim in this sentence does not fit and delete it.